# Based on the Virtual Screening of Multiple Pharmacophores, Docking and Molecular Dynamics Simulation Approaches toward the Discovery of Novel HPPD Inhibitors

**DOI:** 10.3390/ijms21155546

**Published:** 2020-08-03

**Authors:** Ying Fu, Tong Ye, Yong-Xuan Liu, Jian Wang, Fei Ye

**Affiliations:** 1Department of Applied Chemistry, College of Arts and Sciences, Northeast Agricultural University, Harbin 150030, China; fuying@neau.edu.cn (Y.F.); yetong0812@163.com (T.Y.); yongxuan_liu@163.com (Y.-X.L.); 2Key Laboratory of Structure-Based Drug Design & Discovery, Ministry of Education, Shenyang Pharmaceutical University, Shenyang 110016, China

**Keywords:** HPPD inhibitors, pharmacophore model, molecule docking, molecular dynamics, MM/GBSA

## Abstract

4-Hydroxyphenylpyruvate dioxygenase (HPPD) is an iron-dependent non-heme oxygenase involved in the catabolic pathway of tyrosine, which is an important enzyme in the transformation of 4-hydroxyphenylpyruvic acid to homogentisic acid, and thus being considered as herbicide target. Within this study, a set of multiple structure-based pharmacophore models for HPPD inhibitors were developed. The ZINC and natural product database were virtually screened, and 29 compounds were obtained. The binding mode of HPPD and its inhibitors obtained through molecular docking study showed that the residues of Phe424, Phe381, His308, His226, Gln307 and Glu394 were crucial for activity. Molecular-mechanics-generalized born surface area (MM/GBSA) results showed that the coulomb force, lipophilic and van der Waals (vdW) interactions made major contributions to the binding affinity. These efforts will greatly contribute to design novel and effective HPPD inhibitory herbicides.

## 1. Introduction

In recent years, crops severely suffer from infestations of various weeds, and the weeds resistance are increasing seriously [1]. The development of novel green herbicides with highly efficient, eco-friendly, high crop selectivity while low toxicity has become urgent [2,3,4].

4-Hydroxyphenylpyruvate dioxygenase (HPPD) is one of the most important herbicide target enzymes [5]. It is an iron-dependent non-heme oxygenase involved in the catabolic pathway of tyrosine, which catalyzes the transformation of 4-hydroxyphenylpyruvic acid (HPPA) to homogentisic acid (HGA) [6,7,8]. HGA is a key precursor of plastoquinone and vitamin E, and they are essential elements for the light-dependent reaction of photosynthesis. The inhibition of HPPD will lead to photodynamic bleaching of the foliage and ultimately results in necrosis and death [9,10,11,12].

A total of 14 *Arabidopsis thaliana* HPPD (*At*HPPD) were identified, uploaded and stored in the protein database (RCSB PDB), including 11 protein–ligand complexes and 10 resolutions less than three angstroms (Table 1). The 3D structures of HPPD show that all of them share the same 2-His-1-Glu facial triad [13]. In the co-crystallized HPPD inhibitor complexes, Fe^2+^ coordination is achieved by two His amino acids and one Glu amino acid and the β-keto–enol structure of the pyrazole or triketone inhibitors [14,15,16]. The π–π interactions between the phenylalanine and the aromatic moiety of the inhibitors play a key role in the stability of the complex [17,18,19].

Pharmacophore model is a powerful tool for hit lead compounds in medicinal chemistry [20]. Molecular mechanics generalized born surface area (MM/GBSA) is the most well-known method of binding free energy, which was used to assess docking poses, determine structural stability and predict binding affinities [21]. Increasing studies have made major breakthroughs in HPPD design through the use of computational chemistry. A virtual screening based on a pharmacophore model and molecular docking was successfully constructed for screening potential HPPD inhibitors [22,23]. A detailed experimental and computational study have shown that the slow-binding inhibition kinetics of 5CTO, 5DHW and 5YWG, which enabled the rational design of new effective HPPD-targeted drugs or herbicides with longer target residence time and improved performance [24]. Fu et al. successfully constructed a virtual screening based on a pharmacophore model, and then used molecular docking to discern interactions with key residues at the active site of HPPD and identified potential HPPD inhibitors [25]. He et al. summarized the structure of HPPD, inhibitors binding sites, mode of action and their metabolites were [26]. All of these provided clear and solid insights in HPPD and its inhibitors interaction and contributed to design novel and effective HPPD inhibitory herbicides. In this study, a series of structure-based pharmacophore models for HPPD inhibitors were generated and used to virtual screening ZINC and Natural Product database. Molecular docking, molecular dynamics (MD) simulations and molecular-mechanics-generalized born surface area (MM/GBSA) calculation were applicated to illustrate the detailed binding modes of potential inhibitors with HPPD. The virtual filtered strategy is shown in Figure 1.

## 2. Results

### 2.1. Pharmacophores for Virtual Screening

The multiple pharmacophore combination screening method has a higher effective hit rate for active compounds than the single pharmacophore screening. The receptor-based multiple pharmacophore combination screening method is more reliable and helpful for the identification research of potential HPPD inhibitors. The above-mentioned final set pharmacophores were constructed as shown in Figure 2, and the virtual screening process was performed to the 98,133 compounds from ZINC and Natural Product database for 6 times to find new HPPD inhibitors. Finally, 29 compounds were found to be highly fitness with the HPPD based pharmacophore model. In addition, these compounds were optimized for geometry using Schrödinger’s LigPrep module and used for further molecular docking.

### 2.2. Docking Result

Based on the docking results, the ligand benquitrione was completely embedded in the active capsule. benquitrione and amino acids His226, His308, Glu394 coordinated with cobalt ion, meanwhile, the benzene ring of benquitrione formed a face-to-face π–π interaction with Phe381 and hexahydropyrimidine ring formed a vertical π–π interaction with Phe424 (Figure 3A).

Compounds ZINC00004910836, ZINC000040310216 and ZINC000003830381 were found fully embed into the active pocket. Compound ZINC000049180836 not only produced primary metal coordination bonds, moreover, the two benzene rings also interacted with Phe424 and Phe381 via the obvious π–π interactions. Meanwhile, the hydroxyl at benzene ring generated hydrogen bond with Gln307 as depicted in Figure 3B.

The benzene portion of compound ZINC000040310216 formed π–π interactions with Phe424 and Phe381. Compound ZINC000040310216, amino acids His226, His308 and Glu394 all coordinated with cobalt ion. Meanwhile, carbonyl moiety generated hydrogen bond interactions with Gly420 and Asn282 as depicted in Figure 3C.

The docking mode of compound ZINC000003830381 was similar as benquitrione. There was a π–π stacking between cyclopentyl, Phe424 and Phe381. ZINC000003830381 also produced metal coordination bonds together with His226, His308 and Glu394. The hydroxyl on the cyclopentane generated hydrogen bond interactions with Asn423 as displayed in Figure 3D.

Compounds ZINC000035458722, ZINC000012663485 and ZINC000002032320 (Figure 3E–G) also formed π–π interaction with Phe424 and Phe381. The hydroxyl on the aromatic ring part of ZINC000035458722 generated hydrogen bond interaction with Gln307. The pyrrole ring of ZINC000002032320 also generated important hydrogen bond interactions with Phe419. However, none of these three compounds produce metal coordination bonds.

Herein we focused on the three small molecules ZINC000040310216, ZINC000003830381 and ZINC000049180836 because they have shown the function of potential HPPD inhibitors. ZINC000002032320 with the highest docking scores among the left three molecules was also subjected to MD simulations to demonstrate whether the metal coordination was necessary.

### 2.3. MD Simulations

ZINC000049180836, ZINC000040310216 and ZINC000003830381 all formed major ionic interactions with amino acids His226, His308, Glu394, resulting in metal coordination and all the above three compounds inserted in the active pocket between Phe381and Phe424 via π–π interaction (Figure 4A–C). ZINC000049180836 formed three new hydrogen bonds with Gln307, Leu427and Lys421 in HPPD binding site. In HPPD binding pocket, ZINC000049180836 (Figure 7A) displayed crucial interactions with Phe381(0.9), Phe424(0.8), Leu427(0.9) and His226(1.0), His308(1.0), Glu394(1.0) and Gln307(0.9), Lys421(0.8), respectively. Amino acids His226, His308, Glu394 form coordination bonds with metal ions and ligands form π–π interaction with amino acids Phe381 and Phe424.

ZINC000040310216 formed a new hydrogen bond with Gly420 and Asn282 in HPPD binding site (Figure 4B). ZINC000003830381 (Figure 4C) formed important hydrogen bonds with Asn423, Glu394, Gln293 and Gly420 in HPPD binding site. In 5YY6 binding pocket, ZINC000002032320 displayed crucial interactions with Tyr297(a fraction of 0.75), Pro389(0.95), Phe392(0.8) (Figure 4D). The key interactions included hydrogen bonds with Pro389 and Gln307and π–π interaction with Tyr297 for 5YY6, Hydrophobic interaction with Phe392.

ZINC000049180836, ZINC000040310216, ZINC000003830381 and ligand histogram data indicate that these three compounds possess key interactions between ligands and proteins, and both produce important hydrogen bonding properties in drug design. Consequently, the conformations from snapshots were appropriately identical with previous docking modes, which exactly suggested the binding stability.

The RMSD pattern of protein (Cα) and ligand (fitting protein and ligand itself) during the 100 ns simulations is shown in Figure 5. In detail, ‘Lig Fit Protein’ or ligand aligned on protein meant the RMSDs measured the fluctuations of the ligand with respect to the protein [35]. It was obviously that there were some fluctuations with a 3 Å shift for the proteins (HPPD) at the beginning. Hence, the RMSDs fluctuating around some thermal average by approximately 1–3 Å for the entire simulation time exactly illustrated the equilibrated simulation. ZINC000040310216 preserved roughly the same RMSD level with benquitrione, whereas the average values of 5YY6-ZINC000003830381, 5YY6-ZINC000002032320 and 5YY6-ZINC000049180836 were a little higher than that of 5YY6-ZINC000040310216 complex. On the whole, the fluctuations of proteins and ligands attained to be unchanged after 40 ns.

### 2.4. MM/GBSA Calculation

MM/GBSA is a molecular mechanics of binding energy, solvation model, nonpolar solvation term, nonpolar solvent accessible surface area and van der Waals interaction method to define the free energy of binding [35]. MM/GBSA was applied to calculate the binding free energy of the ligand-receptor complexes taking thermodynamic and desolvation parameters into consideration. The Δ*G*_bind_, Δ*G*_bind_ coulomb, Δ*G*_bind_ Hbond, Δ*G*_bind_ lipo and Δ*G*_bind_ vdW parameters involved in the calculation are shown in Table 2. The Glide energy required for compound ZINC00004910836 is the smallest, which was consistent with the previous molecular docking results. Among the calculated parameters of compound ZINC00004910836, the Δ*G*_bind_ value was 55.129 kcal/mol, the Δ*G*_bind_ coulomb value was −48.956 kcal/ mol, the Δ*G*_bind_ Hbond was −0.530 kcal/mol, nonpolar solvation (Δ*G*_bind_ lipo) value was −26.162 kcal/mol and van der Waals (Δ*G*_bind_ vdW) value was −41.324 kcal/mol. From the parameters calculated for each compound, it could be easily concluded that Δ*G* bind coulomb and Δ*G* bind vdW were the main contributors to the binding free energy of compounds. Noticeably, it was also detected that Δ*G* covalent, providing unfavorable energy, was negative to the binding of protein.

## 3. Discussion

Six crystallographic structures of HPPD protein were collected from the RCSB PDB database (PDB ID: 1TG5, 1TFZ, 5YWH, 5YWK, 5YY6 and 5XGK). The interaction between the HPPD protein and potential ligands was calculated using LS. Although these models have similar chemical characteristics, they all describe slightly different interaction patterns that may occur within the HPPD binding site. For example, both the 1TG5 and 1TFZ models contain a hydrogen bond (green) effect with HIS287 and a hydrophobic (yellow) effect with Phe398 and Phe360, but the 1TFZ model has one more aromatic ring effect (blue) than the 1TG5 model, the same as 5YY6. Among the amino acids in the 5YWK and 5YWH models, Phe424, Phe381, Phe392 and Met335 all produced hydrophobic groups (yellow), but the amino acid His308 in 5YWH produced hydrogen bonds (green) as shown in Figure 2.

The co-crystallized ligand benquitrione was redocked into the corresponding 5YY6 protein binding pocket (Figure 6). The RMSD value was 0.55, which confirmed the accuracy and feasibility of the glide docking method.

Flexible docking was performed with the 29 selected candidates against the multiple conformers of the receptor protein 5YY6. Six small molecules with optimal comprehensive conditions were obtained according to the conditions of docking score and the action mode of amino acid. Docking score for the 6 compounds is shown in Table 3. The glide score of the six compounds were lower than others, which indicated that all of them bond with the target protein stable. The Glide energy of ZINC000035458722 is similar as benquitrione. Docking score and Glide gscore are the best criteria for docking situation. The scores of the six compounds are better than the co-crystallized complex ligand benquitrione, indicating that the compounds can complement the protein geometrically and energy well.

Analysis of the cocrystal structures shows some interactions predicted by our ensemble docking model (Figure 7A). Simulation trajectory analysis showed that within this time frame, the RMSD of the ligand changed by about 2 Å relative to the initial (docking) conformation, and the posture was stable. Interactions along the entire trajectories indicate that the interactions with His226, His308, Phe381, Phe424 and Glu394 are conserved and are the residues necessary for HPPD inhibitor binding.

Protein interactions with the ligand were monitored and normalized by a timeline representation over the course of the trajectory (P–L contacts, Figure 7B). The interactions such as–hydrogen bonds (≤2.5 Å), hydrophobic, ionic and water bridge were summarized and categorized by type. In HPPD binding pocket, ligand benquitrione displayed crucial interactions with Phe381(0.9), Phe424(0.8) and His226(1.0), His308(1.0), Glu394(1.0). These results indicated the metal coordination bond interaction was maintained 100% between benquitrione and His226, His308 and Glu394. The interactions with Phe424 and Phe381 hydrophobic bonds are 80% and 90%, respectively.

In conclusion, a reasonable and hierarchical virtual screening process was successfully constructed to identify potential HPPD inhibitors. Molecular docking revealed that compounds ZINC00004910836, ZINC000040310216 and ZINC000003830381 all produced metal coordination with amino acids His226, His308 and Glu394 and formed π–π interactions with amino acids Phe424 and Phe381, and all of them produced more important hydrogen bond interaction with the protein. MD simulation proved the existence of metal coordination bonds and the influence of π–π interaction, which was consistent with the previous molecular docking results. The MD simulation also confirmed that three of the six compounds had high binding efficiency with HPPD. The binding free energy calculated by MM/GBSA demonstrated that the van der Waals (Δ*G*_bind_ vdW) energy term was the major contributor to molecular binding. Overall, the results provide a set of guidelines for design novel and more potent HPPD inhibitors.

## 4. Materials and Methods

### 4.1. Pharmacophore Model Generation and Theoretical Validation

5YWH, 5XGK, 1TFZ, 1TG5, 5YWK and 5YY6 protein–ligand complexes were obtained for a detailed structural analysis after screening the fourteen *At*HPPD protein with the rule of an effective pharmacophore requires 3 or more structural characteristic elements.

Structure-based pharmacophore models for HPPD inhibitors were generated on the 6 HPPD crystal structures using LigandScout (LS, Inte: Ligand, version 4.3, Austria). Then optimize these models to optimize their prediction performance. In the first optimization stage, the size of the features were deleted or adjusted to allow as many active ingredients as possible to draw pharmacophores and exclude inactive compounds according to the contribution map (https://www.mrc-lmb.cam.ac.uk/rajini/index.html) (Figure 8) and the action mode of the amino acids in the PDB [36,37,38]. Features that indicate the interactions with the key residues necessary for inhibition are retained during this process, such as the important interactions with Phe403, Phe398, Phe360, His287 in 1TG5 and 1TFZ. In the second optimization stage, an exclusion volume is added to prevent inactive compounds that are in spatial collision with the protein from being localized by the pharmacophore. This optimization aims to reduce the chances of finding inactive compounds while still correctly identifying the active compounds. Finally, six structure-based pharmacophore models are produced based on 6 PDB proteins combined with the amino acid contribution map and the action mode of the protein ligands in the PDB library.

The 6 pharmacophores obtained above are subjected to multiple virtual screening of 98,133 compounds from the ZINC and Natural Product database. If the number of molecules obtained by the primary screening is too large, the first 200 matching high scores are taken and finally, the selected groups of compounds are summarized to obtain a common intersection compound.

### 4.2. Multiple Receptor Conformers Based Molecular Docking

#### 4.2.1. Ligand Preparation

A total of 29 compounds out of 98,133 were found to be–in high fitness with the pharmacophore model. Using LigPrep, the ionization states, tautomeric forms and 3D conformation of 29 intersection compounds were generated (Schrödinger, LLC, New York, 2018–2). The default conditions were maintained, except for the ionization states which were generated at the target pH of 7.0 ± 0.2 using Epik including the original state [39]. The resulting 3D ligand representations were exported as structure data files, and unique IDs were assigned based on only canonical structures to facilitate post-docking hierarchical data fusion.

#### 4.2.2. Protein Preparation

The protein preparation workflow in Maestro (Schrödinger) was used to preprocess the HPPD protein structure to assign bond sequences and optimize the structure, including hydrogen bond optimization and constrained minimization. Use Schrödinger Prime to add missing side chains when necessary. For each structure, a protein chain with the co-crystal ligand was retained, and more than 5 Å of water molecules were removed from the heteroatom group [40]. The internal hydrogen bond network was optimized, and then the energy minimization was limited.

For the obtained six HPPD protein structures representations, Schrodinger Glide were used to generate docking-grids around the cocrystal binding sites under the default settings. Each cocrystal ligand was re-docked into its corresponding structure validated model; in each case, the cocrystal posture was reproduced with root mean square deviation (RMSD) value <1.5 Å.

#### 4.2.3. Docking Simulation

The docking simulation was performed in Glide. The ligand representation of all compounds obtained through LigPrep was docked with the extra precision (XP) of the default settings of all 6 HPPD models prepared using Glide (Schrodinger), except writing out at most 5 poses per ligand representation and including 32 poses per ligand for post-docking minimization [41]. The obtained 29 small molecules were docked with the above-mentioned six PDB proteins in the glide module of the Maestro-scratch project platform for molecular docking, six compounds were obtained after comparing the conditions of glide gscore, glide energy and the mode of interactions of small molecules with amino acids.

### 4.3. MD Simulation

The complex was introduced into the MD simulations in the Desmond (v3.8) module of Schrödinger software. The simulation system was built in an auto calculated trapezoidal box, which was dissolved with simple point charge (SPC) water and neutralized with appropriate number of counter ions. In addition, the OPLS_2005 force field was used to minimize the energy of complex systems, where the maximum interaction was set to 2000 and convergence threshold was set to 1.0 kcal/mol/Å [42]. The steepest descent and limited memory Broyden–Fletcher–Goldfarb–Shanno (LBFGS) algorithm minimizes the system energy with a maximum of 5000 steps until it reaches a gradient threshold of 25 kcal/mol/Å. An all-atom explicit water MD simulation of 100 ns was performed on four potential small molecules. Each MD system was created from the docking protein–ligand complex corresponding to the best docking score (i.e., the 5YY6 PDB structure and its corresponding docking ligand result in the highest absolute docking score). Finally, the RMSD of the protein backbone and the root mean square fluctuation (RMSF) of the residues were plotted to analyze the structure convergence to equilibrium.

The initial MM/GBSA calculation uses initial and MD to optimize the receptor ligand complex. Use all default parameters, including the VSGB 2.0 solvation model. Calculation of Schrodinger software suite [43,44]. The binding free energy Δ*G*_bind_ was calculated with the MM/GBSA methodology was applied based on stable MD trajectory.
(1)$$\Delta G_{bind}=G_{complex}-G_{receptor}-G_{ligand}$$

*G*_complex_ is the free energy of the complex; *G*_receptor_ is the free energy of the receptor; *G*_ligand_ is the free energy of the ligand.

## Figures and Tables

**Figure 1 ijms-21-05546-f001:**
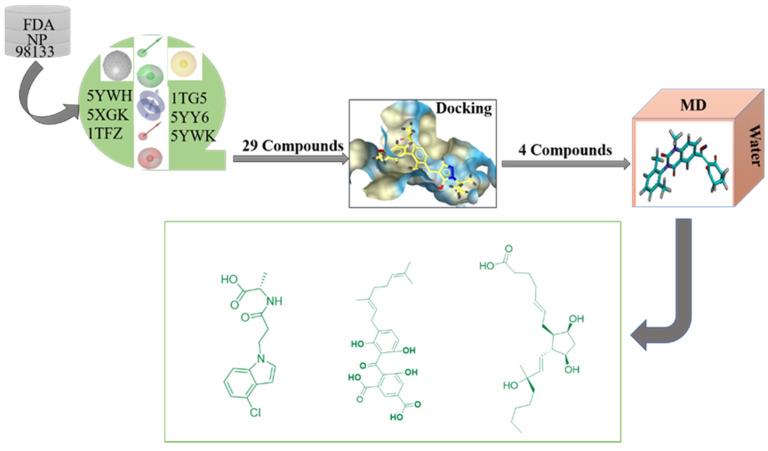
Study design comprising pharmacophore model development, virtual screening, docking and molecular dynamics simulation.

**Figure 2 ijms-21-05546-f002:**
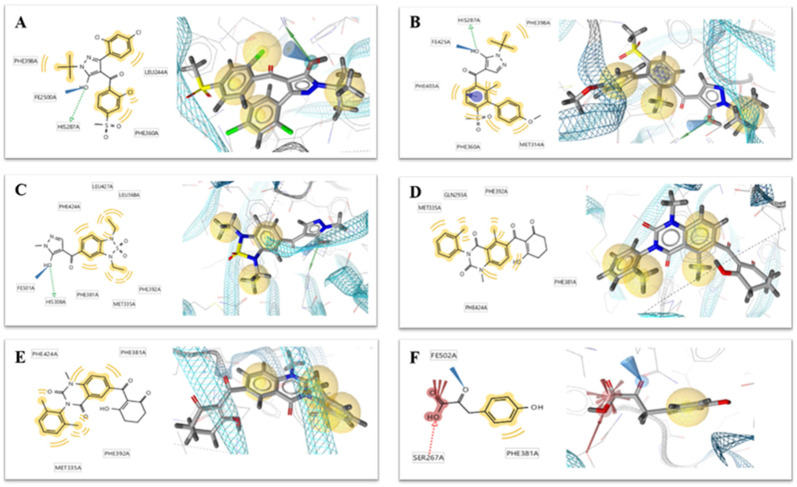
Two-dimensional (left) and 3D (right) charts of structure-based pharmacophore models (**A**) 1TG5, (**B**) 1TFZ, (**C**) 5YWH, (**D**) 5YWK, (**E**) 5YY6 and (**F**) 5XGK for potential 4-hydroxyphenylpyruvate dioxygenase (HPPD) inhibitors in the HPPD binding pocket. The interactions were visualized with the following color code: hydrogen bond acceptor (red arrow), hydrogen bond donor (green arrow), hydrophobic interaction (yellow sphere), aromatic ring feature interaction (purple sphere and arrow).

**Figure 3 ijms-21-05546-f003:**
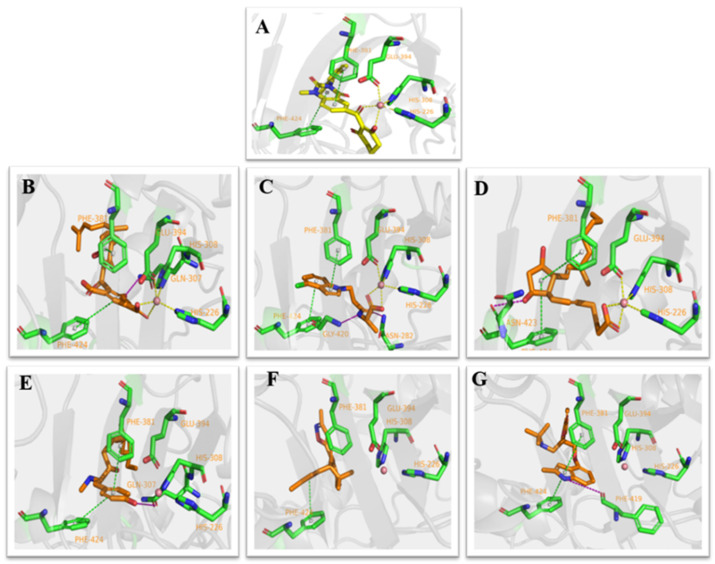
Receptor–ligand interaction of (**A**) benquitrione, (**B**) ZINC000049180836 (**C**) ZINC000040310216, (**D**) ZINC000003830381, (**E**) ZINC000035458722, (**F**) ZINC000012663485 and (**G**) ZINC000002032320 with the HPPD active site.

**Figure 4 ijms-21-05546-f004:**
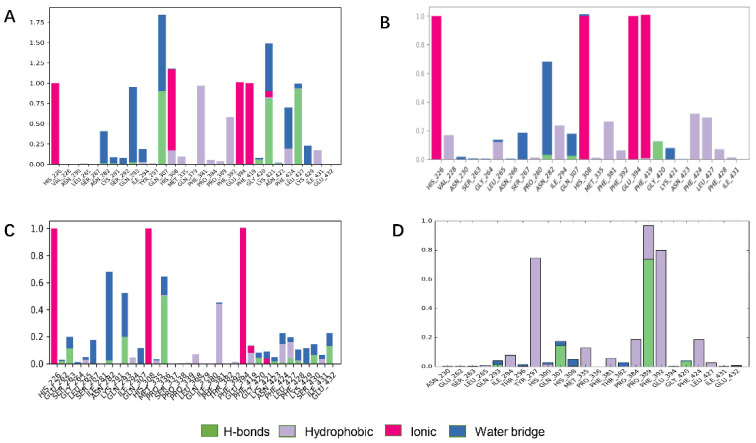
Bar charts of protein–ligand (P–L) contacts. (**A**) ZINC000049180836 (**B**) ZINC000040310216, (**C**) ZINC000003830381, (**D**) ZINC000002032320. Abscissa represents the amino acid number. Ordinate represents interactions fraction.

**Figure 5 ijms-21-05546-f005:**
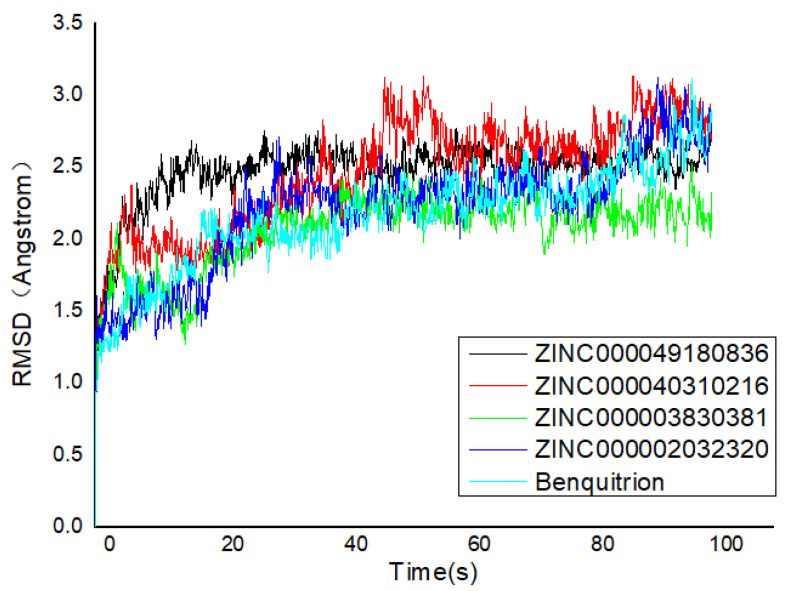
Root mean square deviation (RMSD) trajectories of HPPD–ligand complex during 100-ns simulations.

**Figure 6 ijms-21-05546-f006:**
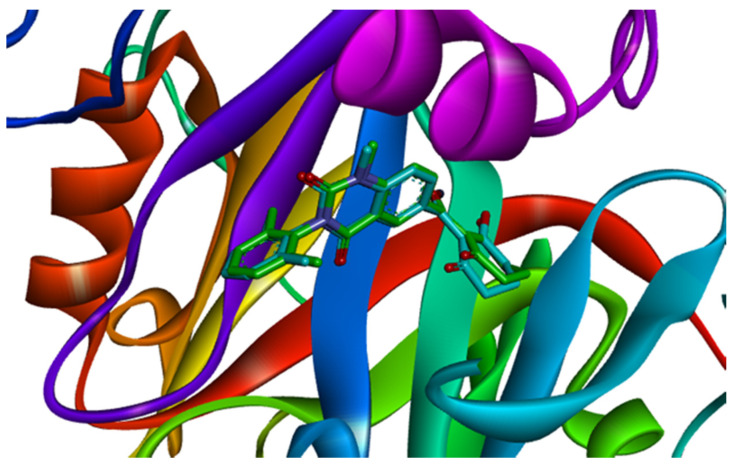
Ligand compared by the glide method. Redocked ligand was green and the native ligand in the crystallographic complex was blue.

**Figure 7 ijms-21-05546-f007:**
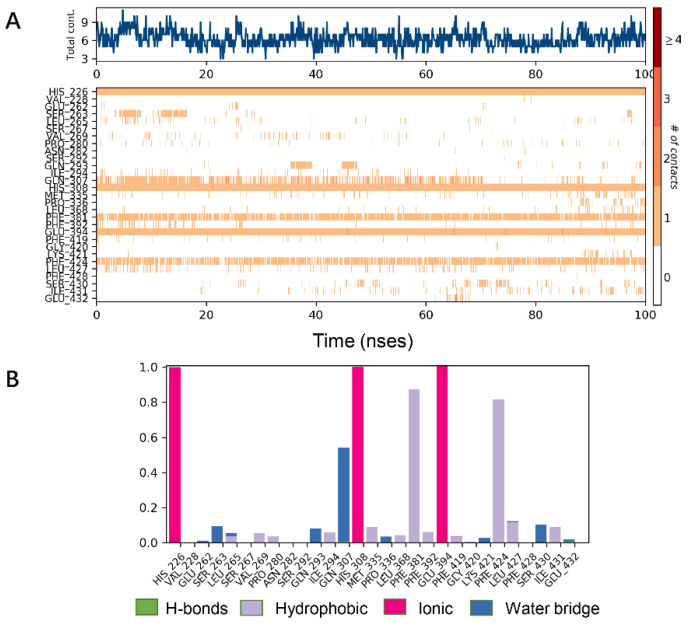
Molecular dynamics (MD) simulations results of 5YY6 protein and its ligand benquitrione. (**A**) Protein–ligand contacts; (**B**) bar charts of protein–ligand (P–L) contacts.

**Figure 8 ijms-21-05546-f008:**
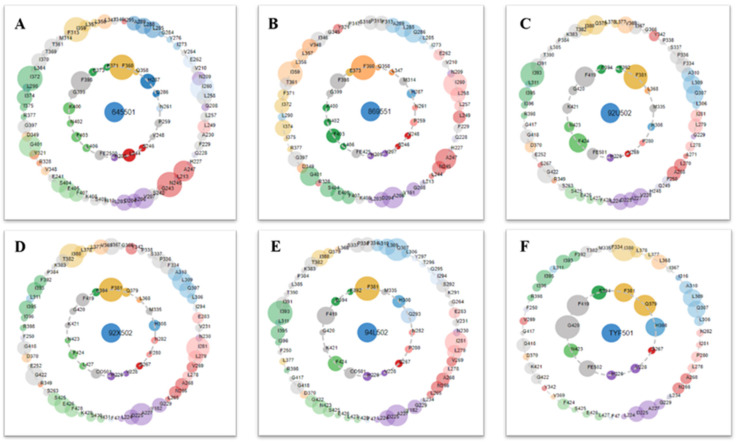
Protein–ligand contacts are represented by asteroid plot. The asteroid plot of *At*HPPD co-crystal structure RCSB PDB database (PDB) code: (**A**) 1TG5, (**B**) 1TFZ, (**C**) 5YWH, (**D**) 5YWK, (**E**) 5YY6 and (**F**) 5XGK). Co-crystal ligand in the center is denoted as blue node. The residues in the inner and outer circle represent directly and indirectly contacting residues, respectively. Size of the circle indicates the number of atomic contacts and the residues are colored according to their secondary structures.

**Table 1 ijms-21-05546-t001:** Structural information of published *Arabidopsis thaliana* HPPD (*At*HPPD) crystals.

Ref.	PDB ID	Resolution (Å)	Ligand	Metal Ion	Released
[27]	1TFZ	1.80	DAS869	Fe	2004
[27]	1TG5	1.90	DAS645	Fe	2004
[27]	1SQD	1.80	-	Fe	2004
[28]	1SP9	3.00	-	Fe	2004
[29]	5XGK	2.60	HPPA	Fe	2018
[30]	6ISD	2.40	sulcotrione	Co	2018
[30]	5YWG	2.60	mesotrione	Co	2019
[30]	6J63	2.62	NTBC	Fe	2019
[31]	5YWH	2.72	Y13508	Fe	2019
[32]	5YWI	2.58	NTBC	Co	2019
[33]	5YWK	2.80	benquitrione-methyl	Co	2019
[30]	5YY6	2.40	benquitrione	Co	2019
[34]	6JX9	1.80	Y17107	Co	2020

**Table 2 ijms-21-05546-t002:** Contributions of various energy components to the binding free energy (kcal mol^−1^).

Compound	Δ*G*_bind_	Δ*G*_bind_ Coulomb	Δ*G*_bind_ Covalent	Δ*G*_bind_ Hbond	Δ*G*_bind_ Lipo	Δ*G*_bind_ vdW
ZINC000049180836	−55.129	−48.956	6.930	−0.530	−26.162	−41.324
ZINC000040310216	−48.339	0.322	8.513	−0.482	−25.319	−48.749
ZINC000003830381	−30.081	−26.801	9.483	−1.109	−27.654	−41.861
ZINC000002032320	−43.629	−15.992	8.294	−0.565	−23.267	−49.487
ZINC000035458722	−50.129	−13.244	3.182	−0.592	−26.8102	−40.3212
ZINC000012663485	−29.341	−45.45	7.568	−0.449	−24.507	−34.352
Benquitrione	−19.652	−27.7484	3.5389	−0.968	−15.086	−32.827

**Table 3 ijms-21-05546-t003:** Two-dimensional structure of the potential HPPD Inhibitors and the evaluation value.

Inhibitors	Structure	Docking Score	Glide Gscore	Glide Energy (kcal mol^−1^)
**ZINC000049180836**	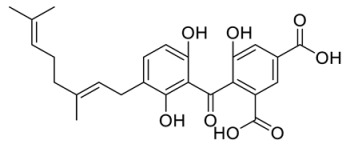	−12.206	−12.294	−52.002
**ZINC000040310216**	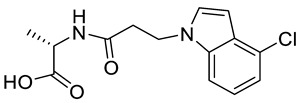	−11.449	−11.449	−47.096
**ZINC000003830381**	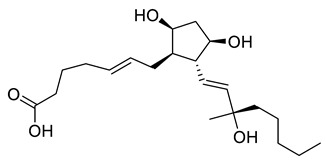	−10.762	−10.765	−31.596
**ZINC000002032320**	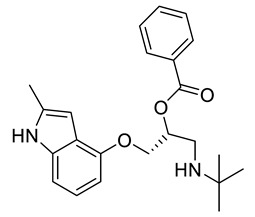	−9.504	−10.859	−30.080
**ZINC000035458722**	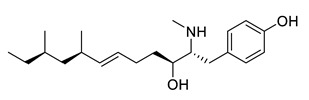	−8.865	−10.903	−51.623
**ZINC000012663485**	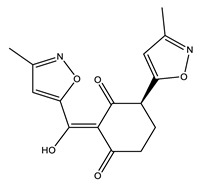	−8.549	−8.549	−32.895
**Benquitrione**	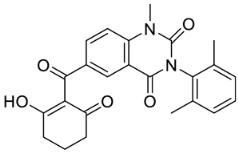	−5.921	−5.921	−44.475

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
