# Peer review of "Based on the Virtual Screening of Multiple Pharmacophores, Docking and Molecular Dynamics Simulation Approaches toward the Discovery of Novel HPPD Inhibitors"

_ijms, 2020, doi:10.3390/ijms21155546_

Round 1

Reviewer 1 Report

A reasonable and hierarchical virtual screening process was successfully
constructed to identify potential HPPD inhibitors.

Interesting study and new simulation/virtual methods to be applied also for other future studies were indicated

The ZINC and Natural Product 17 database were virtually screened and 29 compounds were obtained.

Some recommendations to the authors

-The introduction could be improved indicating also other references for similar studies for discovering novel inhibitors.

-The quality of figure 2 needs to be improved

Reviewer 2 Report

Dear Editor,

Enclosed please find the comments on the following manuscript:

Manuscript ID: ijms-880112

Title: Based on the virtual screening of multiple pharmacophores, docking and molecular dynamics simulation approaches toward the discovery of novel HPPD inhibitors

This article report on multiple structure-based pharmacophore models for HPPD inhibitors and screened Natural Product database for 29 candidates.

The authors made a powerful Pharmacophore model for hit lead compounds in medicinal chemistry.

This is an interesting research to discovery novel green herbicides with highly efficient, eco-friendly, low toxicity to crop.

If authors add the control group of drugs, it will be more helpful to the pharmaceutical industry.